# Exploring the Functionality of Mesh-to-Mesh Value Comparison in Pair-Matching and Its Application to Fragmentary Remains

**DOI:** 10.3390/biology10121303

**Published:** 2021-12-09

**Authors:** Zoe McWhirter, Mara A. Karell, Ali Er, Mustafa Bozdag, Oguzhan Ekizoglu, Elena F. Kranioti

**Affiliations:** 1Edinburgh Unit for Forensic Anthropology, University of Edinburgh, Edinburgh EH8 9AG, UK; zmcwhirter9@gmail.com (Z.M.); ad7900@coventry.ac.uk (M.A.K.); 2School of Psychological, Social and Behavioural Sciences, Faculty of Health and Life Science, Coventry University, Coventry CV1 5FB, UK; 3Department of Radiology, Health Sciences University, Tepecik Training and Research Hospital, Izmir 35180, Turkey; alier1717@yahoo.com (A.E.); bozdagmustafa.84@gmail.com (M.B.); 4Center of Legal Medicine, University of Laussane-Geneva, 1015 Lausanne, Switzerland; drekizoglu@gmail.com; 5Forensic Medicine Unit, Faculty of Medicine, University of Crete, 700 10 Heraklion, Greece

**Keywords:** forensic anthropology, MVC, 3D modelling, pair-matching, computed tomography, fragmentation, clavicle

## Abstract

**Simple Summary:**

Forensic anthropologists often face the task of analysing a mixed group of skeletal remains or matching a solitary bone with the rest of a skeleton to determine if it belongs to the same individual. One of the best ways to do this is by pair-matching left and right bones of the same type. Common pair-matching methods experience issues such as high levels of subjectivity, lack of reliability, or expensive cost of implementation. This study explores the application of the relatively new method, mesh-to-mesh value comparison (MVC), which matches paired bones based on morphological shape to determine the likelihood that they derive from the same individual. This study sought to expand on the success found in past publications using MVC and to see how well it performed on a sample of clavicles, a bone known for having a high degree of bilateral variability, of 80 modern Turkish individuals. This study also explored whether MVC can reliably match fragmented bones to their intact counterpart. Results show MVC successfully matched 88.8% of paired clavicles and suggest the method continues to be a promising avenue for pair-matching that is not affected by ancestry and may be applicable to fragmented remains with further study.

**Abstract:**

Many cases encountered by forensic anthropologists involve commingled remains or isolated elements. Common methods for analysing these contexts are characterised by limitations such as high degrees of subjectivity, high cost of application, or low proven accuracy. This study sought to test mesh-to-mesh value comparison (MCV), a relatively new method for pair-matching skeletal elements, to validate the claims that the technique is unaffected by age, sex and pathology. The sample consisted of 160 three-dimensional clavicle models created from computed tomography (CT) scans of a contemporary Turkish population. Additionally, this research explored the application of MVC to match fragmented elements to their intact counterparts by creating a sample of 480 simulated fragments, consisting of three different types based on the region of the bone they originate from. For comparing whole clavicles, this resulted in a sensitivity value of 87.6% and specificity of 90.9% using ROC analysis comparing clavicles. For the fragment comparisons, each type was compared to the entire clavicles of the opposite side. The results included a range of sensitivity values from 81.3% to 87.6%. Overall results are promising and the MVC technique seems to be a useful technique for matching paired elements that can be accurately applied to a Modern Turkish sample.

## 1. Introduction

Commingled assemblages and isolated skeletal elements are often encountered in the archaeological record as well as in contemporary forensic-related fieldwork [1,2]. The concept of commingled remains refers to a single context in which there is a mixing of fragmented or whole skeletal elements belonging to two or more individuals [3,4]. The definition of commingled assemblages can be further specified as a mixing of the remains to the degree in which further scientific study is necessary to differentiate the various components [4]. The commingled nature of the context can arise through a multitude of processes including animal scavenging, abiotic taphonomic processes, and human activity [4,5,6]. These atypical contexts provide unique challenges in determining the ideal method to sort and analyse the associated osteological material in the pursuit of answering important questions relevant to the study of past populations or forensic investigations [6,7]. Multiple individual burials have often been observed as a regular practice in the Paleolithic period and throughout history [2,4,8,9]. In the more recent past, mass killings have led to many forensic anthropologists encountering an increasing number of sites with commingled remains [4,6]. One of the primary steps for approaching these challenges is to quantify the skeletal elements, define the minimum number of individuals, and then re-associate as many of the skeletal elements as possible in order to individualise the sample [6,8,10].

The most commonly applied method for re-association is the visual examination for similarities in size, shape, and taphonomic changes in order to pair-match skeletal elements [11]. Despite its popularity and longstanding application, there are several limitations to the visual assessment method, most of which stem from the subjective nature inherent in its application. There is no way to standardise observations made by distinct observers and conclusions can be difficult to justify, something that would be a huge detriment to forensic contexts. The accuracy of results is also quite heavily varied as it depends almost entirely on the level of experience held by the individual carrying out the assessment [12].

Another approach often employed for the re-association of human remains is osteometric sorting, which is also concerned with the attempt to pair match left and right skeletal elements. Osteometric sorting can be defined as the “formal use of size and shape to sort bones from one another” [12] (p. 1) and relies upon the metric analysis of different bones and the application of statistical regression formulae to match them with other bones from the same individual [11,13]. The underlying concept is that the degree of robusticity and overall size will be similar amongst all skeletal elements belonging to the same individual. The technique makes an attempt to move beyond the subjective nature of visual assessment by employing statistical models and formulae in order to increase replicability amongst different observers as well as to provide an avenue for quantifying the differences between size and robusticity which would allow for stronger justifications to be made when publishing or presenting resulting pair match conclusions [11,12]. There are many benefits to the technique and include the low cost of utilisation, quick return of results, and low error rates [12]. While it is an improvement upon the previously discussed visual observation method and its heavily subjective nature, there are still many limitations that can be encountered in the use of osteometric sorting. One major limitation is the failure of the method to consider the bilateral asymmetry that may exist within an individual [12,14]. It is well-known that handedness and other factors affect the size and morphology of bones and thus it is erroneous to ignore the effects this asymmetry may have on the expression of robusticity and size within an individual [14]. Another situation in which osteometric sorting may fall short is when attempting to sort individuals of a similar size [12]. This can be a major limitation in a diverse range of settings including, but certainly not limited to, martial-related commingled contexts where most individuals are young adult males from a similar population [9].

While DNA testing is a proven method for re-associating elements, it is also extremely costly and time-consuming and many protocols for dealing with complex commingling include the sorting of remains utilising other less expensive methods prior to the eventual application of DNA analysis, arguably making DNA a last, supplementary step to consider when sorting human remains instead of a primary, stand-alone method [15]. The level of preservation and the degree of taphonomic alterations are additional limitations in the use of DNA analysis for re-associating skeletal elements.

New methodologies employed virtual tools of re-association and pair matching. For example, in a relatively new study, the researchers utilised a sample of 111 metacarpals originating from 17 individuals to perform a pair-matching test. Two-dimensional photographs were utilised to place landmarks on the metacarpals. The hypothesis of the study was that “shape differences would be smaller in bones belonging to the same individual than in those belonging to different individuals” [16] (p. 120). The underlying concepts and theories behind the method are laudable and the consideration of ways in which shape can be quantified is extremely promising and intriguing for the future of pair-matching. Preliminary results showed a range of accurately identified pairs from 75.6% by one observer to 82.9% by the second observer with incorrect pairs made by both [16]. The major limitations of this technique would involve the small simple size, the overall lack of validation studies, the high degree of variability between observer rates of accuracy, and the slightly difficult to reproduce methodology.

Other novel methodologies have focused on the realm of 3D digital analysis in an attempt to overcome the shortcomings of the traditional 2D osteometric sorting method upon which they are based, specifically when applied to high degrees of bilateral asymmetry [17,18]. The first of which utilises digital 3D analysis techniques to compare the 46 variables including dihedral angles, cross-sectional area, and cross-sectional perimeter comparisons. The results showed true positive rates between 0.976 and 1.0 [17]. Similarly, Fancourt et al.’s [18] next-generation osteometric sorting uses 3D computer-automated analysis of data points forming a loop around the perimeter of a bone [18]. The authors found that the 3D analysis outperformed the original 2D osteometric sorting [18]. The promising result from both publications demonstrates the effectiveness of using 3D computerised methods to overcome shortcomings of pre-existing sorting methods.

Recently, a novel virtual method of pair-matching elements in commingled situations was proposed [19]. The mesh-to-mesh value comparison (MVC) method is based on the digital comparison of three-dimensional mesh geometries created from white light-scanned or computed tomography data of skeletal elements. This method has been employed with great success for pair-matching geometries of intact skeletal antimeres, that is, left and right sides, in humeri [19], parietal bones [20], and phalanges [21]. MVC is carried out by comparing the three-dimensional geometry of two skeletal elements and determining a numerical value which demonstrates the amount of similarity of the two elements [19]. The fundamental concept is that two paired elements belonging to the same individual will exhibit greater degrees of similarity than two elements belonging to different individuals. While this concept is not new and is a principal consideration in other pair matching techniques such as osteometric sorting and visual assessment, the traits MVC utilises to determine the similarity between bones is unique. The way the similarity values are generated in MVC is essentially by overlapping two bone models in the same three-dimensional space to determine the places in which the shapes differ and by how much. One of the novel features of MVC is that the method utilises all of the spatial data available and it does so in a three-dimensional landscape. This differs from the other pair-matching methods previously mentioned which focus on characteristics such as size or visual observations as well as from other geometric morphometric methods which rely on a limited number of specific landmarks on the bone as opposed to taking into consideration the entire external morphology and topography of the element in question, as MVC does [16]. MVC uses a “mesh-to-mesh” value which quantifies the difference between two meshes, or models, in millimetres; the lower a mesh-to-mesh value, the more similar the models are. The algorithms utilised to determine a mesh-to-mesh value are based on Iterative Closest Point (ICP) comparison algorithms [19].

Parietal bone pair-matching seemed to be the most successful with 98% sensitivity and 100% specificity [20], followed by the humeri with 100% sensitivity and 60% specificity [19]. Drawbacks on the method include the need for special skills in manipulating 3D data, building 3D models from scans, and securing mesh quality which makes the method time-consuming. Yet, the use of ROC analysis allows the method to be adjusted on the question at hand, that is whether two bones are more likely to belong to the same individual or if excluding that they do is the most probable outcome. This can be achieved by adjusting sensitivity and specificity levels.

Pre-existing methods of re-association commingled remains are varied and diverse. However, many are hindered by limitations such as a high cost of implementation, high degree of subjectivity, low level of accuracy, or a lack of validation studies confirming a proven, replicable accuracy rate of success [4,12,17,18,22,23]. Another important issue is that in many places, there are no available skeletal collections which can be utilised to develop or test these methods for a variety of reasons including ethical concerns, inability to macerate, excavate, or examine remains or due to the lack of documented material. Recently, studies utilising computed tomography (CT) scan data have become more popular and are viewed as a potential solution when physical skeletal material is inaccessible [24,25]. Specifically, there is a current need for techniques which can be accurately applied to the population of Turkey; The Human Rights Association in Turkey produced a report in 2014 discussing the location of 348 mass graves in Turkey containing the remains of 4201 individuals requiring analysis and identification [24] (p. 90). It is especially important that techniques employed by researchers involved in human rights-related excavations worldwide and regardless of time period are as accurate and cross-validated as possible due to the sensitive nature of the investigations. The use of CT scans from the contemporary Turkish population is an ideal approach to solve the current problem concerning the lack of anthropometric data in the country [24].

In this vein, the present study adopted the MVC methodology [19,20,21,26] to investigate its utility in pair-matching clavicles, a paired element that has received limited attention in pair-matching studies. In addition to developing a method for complete clavicles, the study aims to pair-match fragments for the first time, as these can be often encountered in commingled situations. The sample derives from Turkey and the development of a virtual method of pair-matching is an adequate fit for the application in mass graves in the lack of skeletal reference collections in the country as described above.

## 2. Materials and Methods

### 2.1. Sample

For this project, a total of 160 clavicles from randomly selected computed topography (CT) scans taken of 80 individuals were used (Table 1). The CT scan data utilised originates from Tepecik Training and Research hospital in Turkey, were taken in 2016 for a different project, and the files were anonymised prior to receipt by the researcher. The CT scans are in radiological position and were performed using a 64 slice CT scanner (Siemens Solutions, Erlangen, Germany). The scanning parameters are 80 kV, 115 mAs, with a slice thickness of 1 mm and 512 × 512 matrix.

The entire sample included 27 males and 53 females. Ages ranged from 15 to 65 with an average age of 42.5 years. There were eight individuals under the age of 28. The sternal epiphysis of the clavicle does not completely fuse until age 23 for females and 25 for males while visibility of the epiphyseal scar may remain until age 27–29 [27]. Nine of the clavicles in the sample displayed evidence of healed fractures. These were deliberately included in the sample for comparison and results were analysed with the 9 fractured clavicles included as well as with them removed to determine the effect it would have on the attempted pair-matching.

### 2.2. Methods

#### 2.2.1. Segmentation

3D models were created with semi-automated segmentation using the Amira 5.2.2 software package following a modified version of that described by Karell et al. [19] in the first publication of the MVC method. Figure 1 illustrates a model of a left clavicle.

#### 2.2.2. Simulation of Fragments

Following the segmentation process in Amira, the interior of the model was filled using the Fill Holes tool found in the Segmentation Editor. Once this was completed, the models were randomly cropped within the segmentation editor to create three different types of fragments; a fragment of the region adjacent to and including the medial epiphysis, which will be referred to as the sternal fragments, one consisting of portions of the midshaft, referred to as midshaft fragments, and one including the lateral epiphysis which will be referred to as the acromial fragments. This action was carried out for all 160 clavicles to create 160 models of each fragment type (480 in total) as seen in Figure 2.

#### 2.2.3. Mirroring

Following the segmentation and creation of the three-dimensional models, the right clavicles were imported into the Autodesk Netfabb software package and mirrored to create mirrored-rights. This step was carried out to ensure that all models can be appropriately compared once imported into the Viewbox 4.1 beta software.

#### 2.2.4. Alignment

Once all of the right sided models were mirrored, all clavicle models were aligned using the Flexscan 3D software. First, the models were manually aligned as closely as possible. Once they appeared to all be in the same three-dimensional space and orientated in the same direction the alignment and fine-alignment actions were applied to the set. Following this step, the models were exported individually as OBJ files. The purpose for this step in the overall process is to eliminate any three-dimensional distance between the models and serves to reduce the amount of time the alignment step takes during the Viewbox Mesh comparison analysis.

#### 2.2.5. Hollowing

Following the Flexscan alignment process, the models were subjected to a “hollowing” procedure using the Viewbox 4.1 beta software. That is the removal of any internal information and keeping only the external surface data for analysis. The nature of the mesh-to-mesh comparison involves only the morphological shape of the exterior surface of the bone models which makes the internal data irrelevant. Hollowing the models serves the purpose of reducing the amount of data that will need to be processed in the mesh similarity comparison process which will help to reduce the overall computing time. The average amount of data removed from each model was 27%.

#### 2.2.6. Mesh-to-Mesh Value Comparison Using Viewbox

Following the previously described methods for creating and preparing the models, the sample was analysed to generate a mesh comparison value using the Mesh Similarity Tool in the Viewbox 4.1 beta software package. The mesh-to-mesh value is defined as the square root of the mean distances between the points of the two meshes.

The foundational algorithm utilised in the mesh comparison process within Viewbox 4.1 beta is a Trimmed Iterative Closest Point (Trimmed ICP) algorithm. Trimmed ICP has been lauded as a particularly useful moderation of the original ICP which performs well when conditions of three-dimensional comparisons involve the presence of shape defects and measurement outliers [28].

To compare all of the left and right models a folder was created with all models together and selected as the ‘Mesh Folder’. A random model was selected as the reference mesh and the option to ‘compare all meshes in mesh folder to each other’ was chosen. Once all the proper parameters were set the mesh similarity was calculated and completed with a processing time of 21 h and 14 min; however, this time did not require any active input by the user.

Comparisons were carried out for the left and right clavicles, as well as comparisons of the fragmentary models to the complete clavicle models of the opposite side. Once the mesh values were generated for each sample, the generated Excel spreadsheets with the comparison values were used to perform two types of analysis in order to determine sensitivity and specificity values.

### 2.3. Mesh Value Analysis

#### 2.3.1. Lowest Common Value Comparison

The lowest common value comparison method utilises a matrix method for selection in which the lowest mesh-to-mesh values for both the left and right sides must agree in order for a match to be determined. This method was developed by the authors of the original publication about mesh-to-mesh value comparison as an alternative to the previously attempted method of determining a threshold value to use in order to determine matches. As discussed by Karell et al. [19], the use of the cut off threshold value plus two standard deviations did capture almost all of the true matched pairs; however, it also included 51 values that were not true matches. Thus, an improved method for analysis was determined to be necessary [19]. The alternative method was shown to be a better method for selecting true pairs, at least for the humeri in the study. The lowest value comparison method utilises a matrix method for selection in which the lowest mesh-to-mesh values for both the left and right sides must agree in order for a match to be determined. The benefit of this is that there should, in theory, be fewer false matches made.

The process of carrying out lowest value comparison method is executed within Microsoft Excel. This process involves formatting the Viewbox 4.1 beta produced results spreadsheets to determine the lowest three matches for each comparison. Through the use of sorting, macros, and relative references, the lowest agreed upon match by both paired elements is determined and a determination is made whether each row and column match is a true positive, true negative, false positive, or false negative.

A true positive value indicates that the value has been selected as the true match by Viewbox 4.1 beta and is also a known true match based on known sample data. A true negative will be a row in which there are no values selected and there is also no known true match for the model. For the purposes of this study, true negatives were only possible once data were intentionally deleted as original CT scan data were 100% paired. Thus, 20% of the results of each comparison sample were randomly removed to create a portion of true negatives. A false positive is a value in which the comparison method has selected a cell as containing a match but based on previous sample knowledge it is not a true pair. A false negative is when a model is not matched to any other model through the lowest value comparison process but does in fact have a true match.

Following the determination of all rows and columns, all the determinations were used to calculate sensitivity and specificity. Sensitivity was calculated as follows:$$Sensitivity=\frac{True\;Positives}{\left(True\;Positives+False\;Negatives\right)}$$

Specificity was calculated using the following equation:$$Specificity=\frac{True\;Negatives}{\left(True\;Negatives+False\;Positives\right)}$$

#### 2.3.2. Receiver Operating Characteristics (ROC)

A ROC curve is a plot in which the sensitivity is plotted in function of the 100% specificity rate at different cut-off points of a specific parameter [29,30,31]. The plot of a ROC curve allows for the area under ROC curve (AUC) to be calculated. The AUC is a value which measures the success rate a specific parameter has when differentiating between two groups. For the purposes of mesh-to-mesh value comparison, this means that the AUC indicates how well the MVC method would perform with pair-matching. Through the creation of a ROC curve graph, it is possible to determine sensitivity and specificity values. The relationship between sensitivity and specificity is important when it comes to the analysis of ROC curves. A ROC curve of a test which has a perfect discrimination with a sensitivity and specificity of 100% will pass through the upper left corner of the graph.

## 3. Results

A total of 640 models, 160 intact clavicles, and 480 simulated fragments were compared and assessed to determine sensitivity and specificity using both variations of statistical analysis of the MVC method. Results are presented in Table 2.

### 3.1. Lowest Common Value Comparison

#### 3.1.1. Entire Clavicle Models

To determine how well the automated version of the MVC method carried out the pair matching comparison for the clavicle models, two different methods of analysis were performed. The first method of analysing results is known as the lowest value comparison method and was carried out using Microsoft Excel.

This method was developed by the authors of the original publication about mesh-to-mesh value comparison as an alternative to the previously attempted method of determining a threshold value to use in order to determine matches. As discussed by Karell et al. [19], the use of the cut-off threshold value plus two standard deviations did capture almost all of the true matched pairs. However, it also included 51 values that were not true matches, thus an improved method for analysis was determined to be necessary [19]. The alternative method was shown to be a better method for selecting true pairs, at least for the humeri in the study. The lowest value comparison method utilises a matrix method for selection in which the lowest mesh-to-mesh values for both the left and right sides must agree in order for a match to be determined. The benefit of this is that there should, in theory, be fewer false matches made.

The process of carrying out lowest value comparison method is executed within Microsoft Excel. This process involves formatting the Viewbox 4.1 beta-produced results spreadsheets to determine the lowest three matches for each comparison. Through the use of sorting, macros, and relative references, the lowest agreed upon match by both paired elements is determined and a determination is made whether each row and column match is a true positive, true negative, false positive, or false negative.

The analysis of the 160 complete clavicles utilising the lowest common value comparison method yielded a sensitivity of 88.8% and specificity of 42.5% (Table 2).

To determine the impact of pathology and age, separate analyses were performed. A sample of 144 models with the pathological specimens included but the under-28 individuals excluded was analysed and yielded a sensitivity of 81.8% with a specificity of 0% due to the absence of any true negatives. Similarly, a sample of 151 models was analysed with the under-28 clavicles included while excluding the pathological specimens, which yielded a sensitivity of 82.8% and a specificity of 26.1% (Table 2).

#### 3.1.2. Simulated Fragment Models

The comparisons for the sternal fragment type yielded a sensitivity of 65.4% and a specificity of 52.6%. The acromial fragment type produced a sensitivity of 54% and a specificity of 40%. The midshaft fragment type comparison produced a sensitivity value of 31.3% and specificity of 37.8% (Table 2).

### 3.2. ROC Analysis

#### 3.2.1. Entire Clavicle Models

A ROC curve analysis of the data containing the match mesh-to-mesh values for the total sample of 160 entire clavicles produced an AUC value of 0.94 with a standard error of 0.0131 and a *p*-value of <0.0001 (Figure 3a). The sensitivity was 87.6% and the specificity was 90.9% (Table 2).

The ROC analysis of the data containing the entire clavicles with the healed fractures removed yielded an AUC value of 0.953 with a standard error of 0.0106 and a *p*-value of <0.001. The sensitivity was 89.5% and the specificity was 90.1%.

A separate ROC analysis performed on the sample of entire clavicles with the models belonging to individuals under the age of 28 removed produced an AUC of 0.940 with a standard error of 0.0131 and a *p*-value of <0.0001. The sensitivity was 87.6% and the specificity was 90.98% (Table 2).

#### 3.2.2. Simulated Fragment Models

ROC analysis performed on the results of the comparison of left sternal fragments to right clavicles produced an AUC value of 0.895 with a standard error of 0.0150 and a *p*-value of <0.001 (Figure 3b). The sensitivity was 83.8% and the specificity was 83.5%.

ROC analysis performed on the results of the comparison of left midshaft fragments to right clavicles produced an AUC value of 0.848 with a standard error of 0.0162 and a *p*-value of <0.001 (Figure 3c). The sensitivity was 81.3% and the specificity was 74.8%.

ROC analysis performed on the results of the comparison of left acromial fragments to right clavicles produced an AUC value of 0.934 with a standard error of 0.0132 and a *p*-value of <0.001. The sensitivity was 87.5% and the specificity of 87.9%.

Figure 4a illustrates an example of a true match after aligning and comparing the two models (left, mirrored-right) using a colour map. Blue indicates small differences in shape while red indicates large differences. Figure 4b illustrates a mesh-to-mesh comparison of a non-pair.

## 4. Discussion

### 4.1. Comparisons with Other Studies and Methods

The human clavicle is one of the most variable bones in the skeleton in terms of morphological, anatomical, and biomechanical characteristics and has been described as “non-conformist” [27,32,33]. Not only are the clavicles between different individuals extremely diverse but studies have noted a high degree of bilateral asymmetry amongst clavicles belonging to the same individual [27,34]. Clavicles have been extensively studied for several reasons, the most notable being the high rate at which it survives in a good degree of preservation due to the high proportion of compact bone as well as the utility of the medial epiphysis in terms of estimating age at death extending into the third decade of life [27,35,36]. For these reasons, the clavicle was selected to be the focus of this study.

One of the primary aims of this work was to determine the degree of success that can be expected when applying the automated version of mesh-to-mesh value comparison to pair-matching clavicles. The only studies published, to date, on this method (Table 3) are on humeri [19], temporal bones [20], mandibular fossae and condyles [26], and phalanges [21].

When compared to the LCV results of the Karell et al. humeri and temporal studies, the degree of accuracy found in this study is notably lower [19,20] (Table 3). The rate of sensitivity for the automated version of MVC when applied to the sample of 45 humeri is 95% while the resulting sensitivity in this study is 88.8% when analysed with the lowest value comparison (LCV) method. This discrepancy is not wholly unexpected as the clavicle is a much more irregular bone than the humerus and is known for expressing a marked degree of bilateral asymmetry [25,27]. The results are still positive and continue to place the automated mesh-to-mesh value comparison among the more accurate methods for pair-matching.

The 2021 study applying MVC to mandibular condyles and fossae experienced similar results to this study when using LCV analysis, yielding a sensitivity of 88.58% for condyles and 91.17% for fossae. These results are very close to those yielded in the comparisons of 160 clavicles in this study which may suggest that mandibular epiphyses and clavicles both perform similarly in MVC comparisons.

A previous study exploring pair-matching phalanges using the MVC method yielded the most promising and thorough ROC analysis results [21]. In that study, the best pair-matching bone was found to be the proximal phalanx of digit 3 and they found a sensitivity of 87.5% and specificity of 92.4%. This is similar to the 87.6% sensitivity and 90.9% specificity yielded by comparing the entire clavicles in this study.

### 4.2. Analysis Method: LCV vs. ROC

A third primary intention of this study was to explore the differences between the two types of analysis, LCV and ROC, and to determine which performs better when applied to MVC results.

The first type of analysis considered, lowest common value comparison (LCV), has many benefits. The underlying concept is that the lowest match of both the left and right sided models must agree or else it is not determined to be a match. This is especially useful in situations where it is important not to falsely match elements. Additionally, LCV comparison is performed using Microsoft Excel making it a very accessible process as there are no highly specialised software packages which require advanced training or high purchase costs to complete the analysis.

The benefits of the ROC curve analysis are likewise numerous. With the creation of a ROC curve, various types of insight into the data are gained. The additional option to create an interactive dot diagram is extremely useful in certain situations. With the interactive dot diagram, it is possible to choose whether sensitivity or specificity is more important and adjust the threshold in order to determine which values fall below a certain sensitivity-specificity percentage. One situation in which calculating a ROC threshold would be useful is when it is possible to carry out DNA analysis following the MVC process. For example, a mesh-to-mesh value comparison could be undertaken utilizing an interactive ROC dot diagram with 100% sensitivity selected which would mean that the overall number of potential matches would be reduced to those that performed well in the MVC process but with 100% sensitivity, no potential matches would be missed. It would then be simple to perform a DNA analysis on all bones that fall under the line determined by the diagram and then use the results from that analysis to determine the actual true match. This would reduce both the monetary expense as well as the waiting time inherent in the process of carrying out DNA analysis by reducing the original number of elements sent for analysis. This approach would greatly expedite the process as it takes significantly less time to perform an MVC match test than to analyse the DNA of every element in a given assemblage in the pursuit of individualisation.

In addition to the previous benefits, the ROC curve analysis automatically utilises bootstrapping which results in a greater sample size, making accuracy results more reliable [29].Last, the ROC analysis is much less time consuming for the researcher than completing an LCV analysis and is something that could even be put into practice in the field or in situations where spending hours on the computer is not ideal or possible. 

The ideal method for analysing results produced using the automated MVC method cannot be determined without consideration of the type of sample, situation, and expected result of study. In this study, both LCV and ROC performed similar in regard to sensitivity for the entire clavicle models while ROC performed significantly better for the fragment comparisons.

### 4.3. The Effect of Age and Pathology

In addition to the inherent morphology of the clavicle and the effect this may have on the overall success rate of MVC, there are other factors that may have affected the accuracy results in this study. One of these factors is the inclusion of clavicles exhibiting evidence of healed fractures in the sample. In an attempt to determine what effect the presence of these nine healed fractured clavicles may have on the overall study, a separate sample excluding pathological bones was prepared and analysed. The LCV analysis yielded results in which the sample without the fractured clavicles was slightly less sensitive while the ROC analysis produced the opposite results. However, both methods produced similar sensitivity and it can be argued that the difference in accuracy is negligible and thus the presence of healed fractures in the sample was not a major hinderance or a factor that seems to have made a great impact on the overall performance of the MVC method. These results are interesting as they imply that the presence of observable pathology is not something that must be greatly considered when employing the automated version of MVC. Skeletal pathology is a factor that is a common issue for several osteological analysis methods. While the analysis of the effect of pathology in this study was not a key aim and is in no way a conclusive statement on the performance of MVC for pathological samples, these results open an interesting new avenue of future research into the abilities of MVC.

The late-maturing nature of the clavicle is another factor worth consideration when attempting to compare the results of pair-matching clavicles to other studies relying on more typical long or other bones [19,20,21,26]. The sample used in this study included eight individuals under the age of 28 years old. The medial epiphyseal scar can remain visible late into the third decade [27,37]. It is also a commonly reported issue, especially amongst observers with little experience in working with x-rays and CT scans, to miss the medial epiphyseal flake or to be unable to observe the signs of the epiphyseal scar when creating a 3D model of a clavicle [35].

While the clavicle models belonging to these younger individuals were included in the overall sample of 160 clavicles, separate ROC and LCV analyses were completed on a sample with those models in question removed. The results show that when relying on the LCV method of analysis, the inclusion of the clavicles belonging to younger individuals had a positive effect on the results as the sensitivity was approximately 3% greater in the sample where they were included. The results of the ROC analysis were almost identical amongst both the samples indicating that the inclusion of the younger models has essentially no impact when using ROC statistics for analysis. These results could indicate several potential conclusions. It is possible that the errors made during the segmentation process when attempting to observe the flakes or epiphyseal scars were minimal, that the individuals discussed happen to have clavicles that are distinct, and thus pair-matching performs well in their case or, most plausibly, that the sample is too small to have a marked effect on the results. While it is interesting to consider that age is not a factor that negatively affects the MVC process, it should be taken into consideration that a thorough exploration of this concept would require a greater sample of younger individuals.

### 4.4. The Effect of Fragmentation

The exploration of how the automated version of MVC handles the pair matching of fragmentary or incomplete remains was another key aim of this study. Since the application of MVC to incomplete remains has not been thoroughly explored by other researchers to date, that aspect of this study was highly exploratory in nature. A recent study by Acuff et al. applied the MVC method to isolated portions of bone using the mandibular condyles and fossae as a sample [26]. The difference in this study is that the MVC comparisons were made between clavicle fragments and their intact clavicle counterpart as opposed to matching bone fragments to other fragments consisting of the same isolated portion of the entire bone.

The most highly performing fragment type were the fragments which consisted of the area near the lateral epiphysis which are referred to as the acromial fragments in this study. The ROC analysis produced an overall sensitivity of 87.5% and specificity of 87.9% (Table 4) which is only slightly lower than the results of the entire clavicle comparison. The LCV analysis yielded significantly lower results, showing an overall sensitivity of 65.7% and specificity of 40%. These results indicate that the MVC method may have the potential to match fragmentary remains and suggests that future explorations of matching fragments would benefit from focusing on the ROC analysis as it performed significantly better than the LCV. Considering that the fragments are being compared to entire clavicles as opposed to other fragment types, the success is expected to be lower as there is a large portion of the bone which is absent and thus cannot be compared. The potential logic underlying the improved performance of the acromial fragments when compared to the other two types of fragments can be related back to the morphology of the clavicle. Studies have often found this lateral retrocurved section to be one of the most variable regions of the clavicle, making it much more diverse in shape than the midshaft or medial epiphyseal (sternal) sections [25].

The second-best performing fragment type was the medial epiphysis area which, in this study, was referred to as the sternal fragment type. The sternal fragments produced an average LCV sensitivity of 55.4%, specificity of 56.5%, and a ROC sensitivity of 83.8% and specificity 83.5% (Table 4). These results are still positive as even the lower performing LCV method yielded a percentage greater than 50% while the ROC results are more promising.

By far the worst performing fragment type was the ones that are made up on the central aspect of the clavicle and referred to as the midshaft fragments. The resulting ROC analysis yielded a sensitivity of 81.3% and specificity of 74.8% while the LCV yielded a sensitivity of 40.5% and specificity of 37.8%. While the ROC results are still positive, the LCV comparison results are extremely low and less significant than random chance when it comes to determining a true match. The possible logic behind the poor performance of the midshaft fragments is the fact that there is very little variation in shape amongst this area of clavicles. Unlike either epiphyseal area, there are also few notable bony landmarks which aid in the creation of a diverse or unique shape.

## 5. Conclusions

Pair-matching skeletal elements with the goal of re-associating remains to individualise skeletons is one of the most useful approaches to the study of commingled or isolated contexts involving human osteological material. While traditional methodology can be complex and varies greatly between situations, the pre-existing techniques have been proven to be lacking and are often difficult to reproduce between observers or highly dependent on the subjectivity of the researcher. Innovative new methods have investigated the incorporation of machine learning algorithms, computer software, three-dimensional modelling, and increasing utilisation of statistical formulae to combat the issues faced by pre-existing techniques [16,17,18,19,38]. The continual improvement in methodology available for the approach to sorting commingled assemblages is vital as multiple individual contexts are increasingly encountered by both osteoarchaeologists and forensic anthropologists. Thus, the expectation of the degree of accuracy and support for any conclusions made by research carried out in the field of osteological analysis as a whole continues to increase [6].

It is commonly acknowledged that the degree of accuracy involved in creating a biological profile of an individual skeleton tends to be greater when techniques that are either population-specific or shown to be unaffected by ancestral background are employed, making the population-specific validation of methods for analysis exceedingly critical. Through the course of this study, the MVC method for pair matching skeletal elements was analysed and attempts were made to validate its application to a contemporary Turkish sample of 160 three-dimensional clavicle models originating from computed tomography (CT) scans of 80 individuals of mixed age and sex. The overall results did not negate any of the claims made in the original publication and provide further evidence that the MVC process is a promising technique to employ when confronted with large- or small-scale commingled assemblages [19]. Fragmentary remains are often a roadblock when attempting to employ any method of analysis and this study hoped to determine whether that is indeed also the case for MVC. Results were mixed but promising and further research is necessary to determine the degree of accuracy that could be expected when attempting to pair match fragmentary or incomplete remains. This study also provided further support for the continued use of CT scan data as a stand-in for physical skeletal collections when necessary and the positive effect this can have on the validation of methods for specific populations lacking in skeletal material available for research purposes.

## Figures and Tables

**Figure 1 biology-10-01303-f001:**
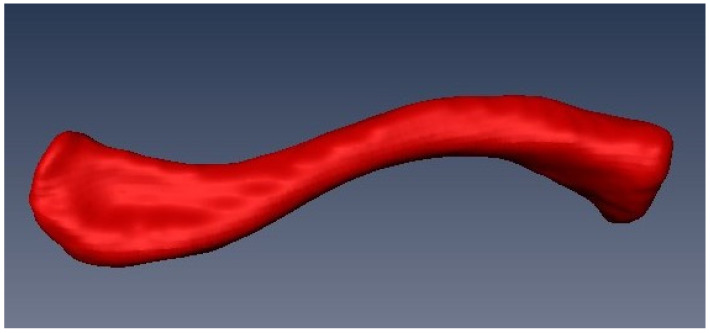
A completed model of the left clavicle belonging to individual 82. Created with Amira 5.2.2.

**Figure 2 biology-10-01303-f002:**
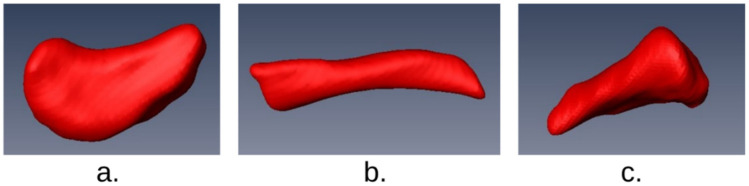
Examples of simulated fragments created using Amira 5.2.2.: (**a**) acromial fragment type, (**b**) midshaft fragment type, (**c**) sternal fragment type.

**Figure 3 biology-10-01303-f003:**
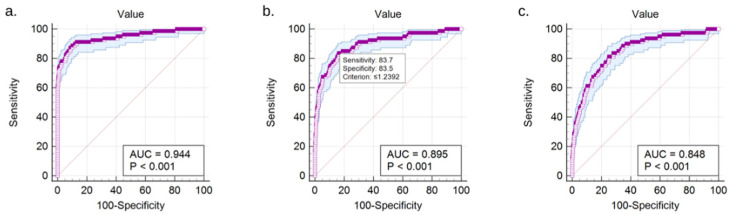
ROC curve diagram for (**a**) total sample of 160 clavicle models, (**b**) left sternal fragments matched to right entire clavicle models, and (**c**) left midshaft fragments matched to right entire clavicle models.

**Figure 4 biology-10-01303-f004:**
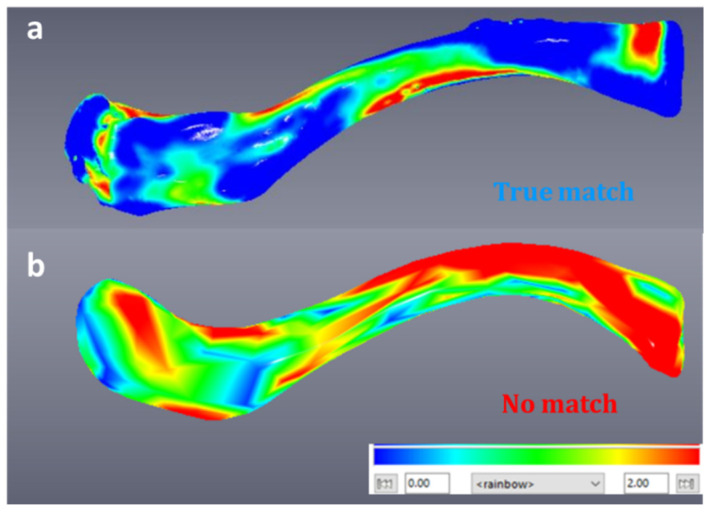
(**a**) True match of left and mirrored-right clavicle-visualisation of shape differences using a colour map in Viewbox beta software. (**b**) Visualisation of shape differences using a colour map for a left and mirrored-right clavicle that are not a true match.

**Table 1 biology-10-01303-t001:** Biological information of the Modern Turkish sample used in this study.

Sex	Number (Total)	Healed Fractures	Under 28 Years
Male	54	4	6
Female	106	5	10
Total	160	9	16

**Table 2 biology-10-01303-t002:** Results of all comparisons analysed in this study using both LCV and ROC statistical methods.

	LCV		ROC	
	Sensitivity	Specificity	Sensitivity	Specificity
160 clavicles	88.8%	42.5%	87.6%	90.9%
151 clavicles (Pathological specimens excluded)	82.8%	26.1%	89.5%	90.1%
144 clavicles (Under age 28 excluded)	81.8%	0%	87.6%	90.98%
160 acromial fragments	54%	40%	87.6%	87.9%
160 midshaft fragments	31.3%	37.8%	81.3%	74.8%
160 sternal fragments	65.4%	52.6%	83.8%	83.5%

**Table 3 biology-10-01303-t003:** LCV results of this study compared to previous MVC publications by Karell et al. (2016, 2017) and Acuff et al. (2021).

Sample	Author	Sensitivity	Specificity
45 mixed ancestry humeri (24 individuals)	Karell et al., 2016	95%	60%
120 Modern Greek temporals (60 individuals)	Karell et al., 2017	98%	100%
70 Cretan mandibular condyles (35 individuals)	Acuff et al., 2021	88.58%	0%
69 Cretan mandibular fossae (35 individuals)	Acuff et al., 2021	91.17%	100%
160 Modern Turkish clavicles (80 individuals)	This study	88.8%	42.5%
160 acromial fragments (80 individuals)	This study	54%	40%
160 midshaft fragments (80 individuals)	This study	31.3%	37.3%
160 sternal fragments (80 individuals)	This study	65.4%	52.6%

**Table 4 biology-10-01303-t004:** Summary table of MVC fragment comparison results.

Fragment Type	LCV Sensitivity	LCV Specificity	ROC Sensitivity	ROC Specificity
Sternal	55.4%	56.5%	83.8%	83.5%
Midshaft	40.5%	37.8%	81.3%	74.8%
Acromial	65.7%	40%	87.5%	87.9%
Entire clavicles	88.8%	42.5%	87.6%	91.1%

## Data Availability

Data can be made available upon reasonable request.

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
