# Peer review of "Exploring the Functionality of Mesh-to-Mesh Value Comparison in Pair-Matching and Its Application to Fragmentary Remains"

_biology, 2021, doi:10.3390/biology10121303_

Round 1
Reviewer 1 Report
This paper builds off of the work by Karell and colleagues to evaluate how well the MVC approach works on complete and fragmentary claviculae. The use of simulated fragmentary elements is scientifically sound and will provide useful information for future research. Fragmentary elements are often not included in research on osteometric sorting, and it is good to see this paper deals with this issue. However, there are some key components missing from this paper that prevent it from being digestible for the majority of readers who aren’t familiar with the previous work cited in the paper.
I would also recommend providing a more updated literature review in regard to three-dimensional pair-matching. There have been two relatively new publications that focus specifically on this topic:
Advances in Osteometric Sorting: Utilizing Diaphyseal CSG Properties for Lower Limb Skeletal Pair-Matching (https://onlinelibrary.wiley.com/doi/10.1111/1556-4029.14480)
Next-generation osteometric sorting: Using 3D shape, elliptical Fourier analysis, and Hausdorff distance to optimize osteological pair-matching (https://onlinelibrary.wiley.com/doi/10.1111/1556-4029.14681)
Key Issues:
- It would be helpful to readers to provide exactly how sensitivity and specificity are calculated. What the true positive and true negative rates refer to entirely depends on whether the method excludes or includes matches. Providing clarification on this would be useful.
- A more technical overview of the previous MVC approach would also be beneficial to readers who may not be familiar with the original publication by Karell and colleagues.
- Specifically, I would like to see a mathematical overview of how the mesh-to-mesh value is calculated. As far as I can tell, this terminology is not standard within the three-dimensional literature, and it could refer to several different types of distance metrics.
- If I am recalling correctly, I believe the original lowest common value approach by Karell utilized the lowest agreed upon distances within the first lowest three distances. Am I correct in assuming this paper looks at more than just the lowest three distances to identify this agreement?
- I am confused by the ROV curve analysis in this paper. You correctly state that it provides a measure of success rate of a specific parameter, which is correct. It shows how the success changes as that parameter is adjusted. However, nowhere in the paper do you tell us what that parameter is. It seems that this parameter is in essence a new approach for identifying matches and the paper would greatly benefit from an explanation of how this works. Is it based on a short list approach or something similar?
- I see in the discussion that you list the ROC curve uses bootstrapping. However, that would imply that the sensitivity and specificity obtained are extrapolated and not always reflective of how it would perform on a single isolated commingled forensic case. Are you sure you aren’t overfitting the results by using bootstrapping?
- It says that 20% of the results of each comparison sample were randomly removed to create a portion of true negatives for the LCV approach. Could you provide further explanation on why this is required?
Author Response
Reviewer 1
Recommendation: Revise background info with newer articles about 3-D pair matching
Suggested articles
Advances in Osteometric Sorting: Utilizing Diaphyseal CSG Properties for Lower Limb Skeletal Pair-Matching (https://onlinelibrary.wiley.com/doi/10.1111/1556-4029.14480)
Next-generation osteometric sorting: Using 3D shape, elliptical Fourier analysis, and Hausdorff distance to optimize osteological pair-matching (https://onlinelibrary.wiley.com/doi/10.1111/1556-4029.14681)
Response: We have incorporated the studies on page 3
Other novel methodologies have focused on the realm of 3D digital analysis in an attempt to overcome the shortcomings of the traditional 2D osteometric sorting method upon which they are based, specifically when applied to high degrees of bilateral asymmetry [37, 38]. The first of which utilizes digital 3D analysis techniques to compare the 46 variables including dihedral angles, cross-sectional area and cross-sectional perimeter comparisons. The results showed true positive rates between 0.976 and 1.0 [37]. Similarly, Fancourt et al.’s [38] next-generation osteometric sorting uses 3D computer-automated analysis of data points forming a loop around the perimeter of a bone [x]. The authors found that the 3D analysis outperformed the original 2D osteometric sorting [38]. The promising result from both publications demonstrates the effectiveness of using 3D computerized methods to overcome shortcomings of pre-existing sorting methods.
Key Issues:
Issue 1: “It would be helpful to readers to provide exactly how sensitivity and specificity are calculated. What the true positive and true negative rates refer to entirely depends on whether the method excludes or includes matches. Providing clarification on this would be useful.”
Response: We added the following information in page 7
A true positive value indicates that the value has been selected as the true match by Viewbox 4.1 beta and is also a known true match based on known sample data. A true negative will be a row in which there are no values selected and there is also no known true match for the model, for the purposes of this study true negatives were only possible once data was intentionally deleted as original CT scan data was 100% paired. A false positive is a value in which the comparison method has selected a cell as containing a match however based on previous sample knowledge it is not a true pair. A false negative is when a model is not matched to any other model through the lowest value comparison process but does in fact have a true match.
Following the determination of all rows and columns, all the determinations will be used to calculate sensitivity and specificity. Sensitivity will be calculated as follows:
Specificity will be calculated using the following equation:
Issue 2: “A more technical overview of the previous MVC approach would also be beneficial to readers who may not be familiar with the original publication by Karell and colleagues.”
Response: More MVC overview was added into the introduction section and throughout the methodology inn Introduction (page 3)
MVC is carried out by comparing the three-dimensional geometry of two skeletal elements and determining a numerical value which demonstrates the amount of similarity of the two elements [17]. The fundamental concept is that two paired elements belonging to the same individual will exhibit greater degrees of similarity than two elements belonging to different individuals. While this concept is not new and is a principle consideration in other pair matching techniques such as osteometric sorting and visual assessment, the traits MVC utilizes to determine the similarity between bones is unique. The way the similarity values are generated in MVC is essentially by overlapping two bone models in the same three-dimensional space to determine the places in which the shapes differ and by how much. One of the novel features of MVC is that the method utilizes all of the spatial data available and it does so in a three-dimensional landscape. This differs from the other pair-matching methods previously mentioned which focus on characteristics such as size or visual observations as well as from other geometric morphometric methods which rely on a limited number of specific landmarks on the bone as opposed to taking into consideration the entire external morphology and topography of the element in question, as MVC does [16].
Issue 3: “Specifically, I would like to see a mathematical overview of how the mesh-to-mesh value is calculated. As far as I can tell, this terminology is not standard within the three-dimensional literature, and it could refer to several different types of distance metrics.”
Response:
Mathematical overview was provided in the introduction section and expanded upon in section 2.2.6
Introduction: MVC uses a “mesh-to-mesh” value which quantifies the difference between two meshes, or models, in millimeters, the lower a mesh-to-mesh value, the more similar the models are. The algorithms utilized to determine a mesh-to-mesh value are based on Iterative Closest Point (ICP) comparison algorithms [17]
Methods: 2.2.6: The foundational algorithm utilized in the mesh comparison process within Viewbox 4.1 beta is a Trimmed Iterative Closest Point (Trimmed ICP) algorithm. Trimmed ICP has been lauded as a particularly useful moderation of the original ICP which performs well when conditions of three-dimensional comparisons involve the presence of shape defects and measurement outliers [36].
Issue 4: “If I am recalling correctly, I believe the original lowest common value approach by Karell utilized the lowest agreed upon distances within the first lowest three distances. Am I correct in assuming this paper looks at more than just the lowest three distances to identify this agreement?”
It is the same process just more automated. A detailed description can be found in section 2.3.1
Issue 5: “I am confused by the ROC curve analysis in this paper. You correctly state that it provides a measure of success rate of a specific parameter, which is correct. It shows how the success changes as that parameter is adjusted. However, nowhere in the paper do you tell us what that parameter is. It seems that this parameter is in essence a new approach for identifying matches and the paper would greatly benefit from an explanation of how this works. Is it based on a short list approach or something similar?”
The parametric is the mesh-to-mesh value. More information was added into the methods and discussion sections about the ROC Curve analysis and how the graph can be used for analysis:
Page 7: Through the creation of a ROC Curve graph it is possible to determine sensitivity and specificity values. The relationship between sensitivity and specificity is important when it comes to the analysis of ROC curves. A ROC curve of a test which has a perfect discrimination with a sensitivity and specificity of 100% will pass through the upper left corner of the graph.
Page 11: The benefits of the ROC Curve analysis are likewise numerous. With the creation of a ROC Curve numerous types of insight into the data are gained. The additional option to create an interactive dot diagram is extremely useful in certain situations. With the interactive dot diagram it is possible to choose whether sensitivity or specificity is more important and adjust the threshold in order to determine which values fall below a certain sensitivity-specificity percentage. One situation in which calculating a ROC threshold would be useful is when it is possible to carry out DNA Analysis following the MVC process. For example, A mesh-to-mesh value comparison could be undertaken utilizing an interactive ROC dot diagram with 100% sensitivity selected which would mean that the overall number of potential matches would be reduce to those that performed will in the MVC process but with 100% sensitivity, no potential matches would be missed. It would then be simple to perform a DNA analysis on all bones that fall under the line determined by the diagram and then use the results from that analysis to determine the actual true match. This would reduce both the monetary expense as well as the waiting time inherent in the process of carrying out DNA Analysis by reducing the original number of elements sent for analysis which would greatly expedite the process as it takes significantly less time to perform an MVC match test than to analyse the DNA of every element in a given assemblage in the pursuit of individualization.
Issue 6: “I see in the discussion that you list the ROC curve uses bootstrapping. However, that would imply that the sensitivity and specificity obtained are extrapolated and not always reflective of how it would perform on a single isolated commingled forensic case. Are you sure you aren’t overfitting the results by using bootstrapping?”
Response: The bootstrapping is used to confirm the results on the original sample not to over fit the results but the contrary, to make sure that the results are not over fitted using a specific sample. This is a common procedure and in our case it is not affecting the reported results.
Issue 7: “It says that 20% of the results of each comparison sample were randomly removed to create a portion of true negatives for the LCV approach. Could you provide further explanation on why this is required?”
Response:
20% of the matches were randomly removed to allow for the existence of true negatives which would not have occurred naturally due to the nature of the sample being from CT scans of individuals with all elements being naturally paired. This has been clarified in the methods section, as seen below in page 7:
“A true negative will be a row in which there are no values selected and there is also no known true match for the model, for the purposes of this study true negatives were only possible once data was intentionally deleted as original CT scan data was 100% paired.”
Thank you very much for the time to review our manuscript and your comments
Sincerely
The authors
Reviewer 2 Report
Dear Authors, I find this article to be well-written and highly relevant to the field of anthropology and mass graves. Although the clavicle is a tricky bone to work with in terms of its variability, you managed to explore a method that will not easily be outperformed by another. I enjoyed reading your work, and would like to see it published. It is, however, unfortunate that this technology and expertise are not available to all forensic facilities dealing with comingled remains. The draft does not need any large changes. Congratulations on doing a good job!
Additional Comments:
1. What is the main question addressed by the research?
The researchers aimed at providing an improved method for matching individual clavicles/bones in cases where comingled remains are found. In order to sort such remains in a manner as to determine how many individuals are present, a valid method is often needed. The authors explored such a method, and succeeded in explaining the pro's and con's when employing 3D software techniques for this purpose.
2. Do you consider the topic original or relevant in the field? Does it
address a specific gap in the field?
Yes, definitely relevant. It provides a more scientific way of sorting remains, whereas in some cases, bones are often paired by guessing (looking at the size and shape), which is way too subjective.
3. What does it add to the subject area compared with other published
material?
The article explains a more advanced method for pairing bones clearly, that may be used at any facility with the relevant equipment, software and expertise.
4. What specific improvements should the authors consider regarding the
methodology? What further controls should be considered?
None. Methodology is sufficient.
5. Are the conclusions consistent with the evidence and arguments
presented and do they address the main question posed?
Yes.
6. Are the references appropriate?
Yes.
7. Please include any additional comments on the tables and figures.
Tables and figures are sufficient. If I have to be picky about it, I would suggest that Figures 1 and 2 use a different colour to red for visualization, as reg sometimes appears harsh to the eye. However, this often depends on the software used.
Author Response
Thank you very much for your time to review our manuscript and your encouraging comments!!!
Round 2
Reviewer 1 Report
Thank you for taking the time to consider my review comments. The clarifications and edits have definitely improved the overall paper.